# Genetic effect on free amino acid contents of egg yolk and albumen using five different chicken genotypes under floor rearing system

Kenji Nishimura[1], Daichi Ijiri[2], Saki Shimamoto[2,3], Masahiro Takaya[1,4], Akira Ohtsuka[2], Tatsuhiko Goto[1,5]*

1 Department of Life and Food Science, Obihiro University of Agriculture and Veterinary Medicine, Obihiro, Hokkaido, Japan, 2 Department of Biochemical Science and Technology, Kagoshima University, Korimoto, Kagoshima, Japan, 3 Graduate School of Science and Technology, Niigata University, Niigata, Japan, 4 Hokkaido Tokachi Area Regional Food Processing Technology Center, Tokachi Foundation, Obihiro, Japan, 5 Research Center for Global Agromedicine, Obihiro University of Agriculture and Veterinary Medicine, Obihiro, Hokkaido, Japan

* tats.goto@obihiro.ac.jp

**Data Availability Statement:** All relevant data are within the manuscript and its Supporting Information files.

## Abstract

Chicken eggs play an important role as food resources in the world. Although genetic effects on yolk and albumen contents have been reported, the number of chicken genotypes analyzed so far is still limited. To investigate the effect of genetic background on 10 egg traits, 19 yolk amino acid traits, and 19 albumen amino acid traits, we evaluated a total of 58 eggs from five genotypes: two Japanese indigenous breeds (Ukokkei and Nagoya) and three hybrids (Araucana cross, Kurohisui, and Boris Brown) under a floor rearing system. One-way ANOVA revealed significant effects of genotype on 10 egg traits, 8 yolk amino acids (Asp, Glu, Ser, Gly, Thr, Tyr, Cys, and Leu), and 11 albumen amino acids (Asp, Glu, Asn, Ser, Gln, His, Ala, Tyr, Trp, Phe, and Ile) contents. Moderate to strong positive phenotypic correlations among traits within each trait category (size and weight traits, yolk amino acid traits, and albumen amino acid traits), whereas there were basically no or weak correlations among the trait categories. However, a unique feature was found in the Araucana cross indicating moderate positive correlations of amino acids between yolk and albumen. These results suggest that genetic factors can modify not only the size and weight of the egg and eggshell color but also yolk and albumen free amino acids contents.

## Introduction

Chicken is likely to be domesticated from red jungle fowls in southwestern China, Thailand, and Myanmar around 9500 ± 3300 years ago [1]. The number of chickens exceeded 22 billion in 2017, while there are approximately 1600 local chicken breeds around the world [2]. Many kinds of chickens have extreme phenotypic characteristics such as body size, feathers, growth, eggs, meat, and behavior. For example, the Japanese indigenous chicken breeds of "Onaga-dori" has extremely long tail feathers in male [3,4], while the weight of male "Oh-Shamo"

**Funding:** This study was supported in part by Grants-in-Aid for the Regional R&D Proposal-Based Program from Northern Advancement Center for Science & Technology of Hokkaido Japan, the Kieikai Research Foundation, and Obihiro City-Obihiro University of Agriculture and Veterinary Medicine Collaboration Project. The funders had no role in study design, data collection and analysis, decision to publish, or preparation of the manuscript.

**Competing interests:** The authors have declared that no competing interests exist.

reaches 4–6 kg [3,5,6]. Thus, the indigenous chickens have a large potential for studying genetic and phenotypic diversity [7].

Eggs are very useful sources of high-quality protein in developed and developing countries alike [8]. From the first half of the 20th century, specialized breeding towards high egg production was started to increase the laying performance. As a result of the breeding using phenotypic and pedigree information, hens of recent chicken lines express high egg production with peak egg-laying rates of 95–97% [9]. However, the commercial lines based on White Leghorn, White Plymouth Rock, and Rhode Island Red (RIR) in the layer industry are only a handful of breeds as opposed to indigenous breeds of chickens in the world. To improve the existing egg quality traits, further investigating these genetic resources may be important.

Egg traits such as egg weight, eggshell color, yolk size, albumen height [10–14], and ingredients of yolk and albumen [15–17] are fairly influenced by genetic factors. Therefore, eggs from several breeds may express a large variation in the appearance of eggs and contents of yolk and albumen caused by differences in genetic background. For example, the eggshell color of the Araucana that has South American origin is blue, which is significantly different from the white eggshell color of White Leghorn [18,19]. Also, it has been reported that the contents of fatty acids, cholesterol, protein, ash, and mineral in eggs [20–22] significantly differed among several breeds and strains. Significant effects of breed differences on free amino acids of yolk and albumen have been reported [17,23–25]. Free amino acids are well known to act on food tastes [26].

The relationship between amino acids and taste (sweetness, sourness, saltiness, bitterness, and umami) has been studied. It is known that amino acids induce the response of taste-related sensory cells using a mouse model [27]. Glutamic acid has an effect towards umami [28], whereas serine is known to be related to sweetness [26]. Goto et al. [25] reported significant breed and feed effects on amino acids and taste sensor traits of albumen and yolk using Australorp and RIR fed mixed and fermented feeds. Although Goto et al. [17] reported genetic impact on free amino acids of yolk and albumen using Australorp (AUS), Nagoya (NGY), RIR, Shamo (SHA), Ukokkei (UKO), and two $F_1$ hybrids (NGY × RIR and SHA × RIR), the number of breeds analyzed is still limited. Moreover, these studies have been conducted under a cage rearing system. Therefore, we hypothesized that there are some different features in free amino acid contents of egg yolk and albumen using untapped several chicken genotypes under floor rearing system. The purpose of this study was to investigate the effect of genetic background on egg traits and free amino acid contents of yolk and albumen using five genotypes, Araucana cross (ARA), Kurohisui (KRH), UKO, NGY, and a representative Brown layer (Boris Brown; BOR), under floor rearing system.

## Materials and methods

This study was approved by the Experimental Animal Committee of the Obihiro University of Agriculture and Veterinary Medicine (authorization number 18–15).

### Chickens

A total of 58 eggs obtained from adult hens (72 weeks of age) from ARA (n = 10), KRH (n = 12), UKO (n = 12), NGY (n = 12), and BOR (n = 12) were used. Although UKO and NGY are pure Japanese indigenous breeds, ARA, KRH, and BOR are hybrids based on crosses of two or more breeds. Since BOR (GHEN Corporation, Japan) is the world's brown egg layer (Hy-Line Brown; Hy-Line International, USA), we selected BOR as representative hen for the comparison. Five genotypes of chickens in the Tago Poultry Farm (Obihiro, Japan) were kept under a floor rearing system. The hens were reared under the 16 h light and 8 h dark

photoperiod cycle with free access to feeds and water. Since we collected all eggs on the day of July, a total of 58 hens were used in this study. The Tago Poultry Farm uses the mixed feed for layers (16.5% CP, 2770 ME kcal/kg; Chubu Shiryo Co., Ltd., Japan) for all the genotypes.

## Measurement of egg traits

We measured a total of 10 egg traits including egg weight (EW), yolk weight (YW), albumen weight (AW), eggshell weight (SW), length of the long axis of egg (LLE), length of the short axis of egg (LSE), eggshell thickness (ST), eggshell color lightness (SCL), eggshell color redness (SCR), and eggshell color yellowness (SCY). Egg weight, yolk weight, and eggshell weight were measured by an electronic balance (EK-6000H; A&D Company, Ltd., Japan), whereas albumen weight was obtained by subtracting weights of yolk and eggshell from egg weight. Lengths of the long and short axes of egg were measured using a digital caliper (P01 110–120; ASONE, Japan), while a Peacock dial pipe gauge (P-1; Ozaki MFG Co., Ltd., Japan) was used to measure the eggshell thickness. Eggshell colors in units of the CIE L*a*b* color space were measured by a chromameter (CR-10 Plus Color Reader; Konica Minolta Japan, Inc., Japan), which confirmed that breeders can use Minolta colorimetry with confidence to assess differences in staining to estimate cuticle deposition (absorbance at 640 nm) [11]. After measuring the 10 egg traits, the yolk was diluted 2-fold with distilled water (DW) for free amino acid analysis. Then, the yolk solution and albumen were stored at -30˚C until use.

## Free amino acid analysis of yolk

The 5 mL of 2-fold diluted yolk solution was thawed with lukewarm water. Then, 5 mL of 16% trichloroacetic acid solution (FUJIFILM Wako Chemicals, Japan) was added to each sample and mixed thoroughly by a vortex. The mixture was centrifuged at $1,400 \times g$ for 15 minutes using a table-top centrifuge (model 2410; KUBOTA Corporation Co., Ltd., Japan) or a centrifuge (RX II series; HITACHI Ltd., Japan). The supernatant was filtered using a syringe (NIPRO Corporation, Japan) and a 0.45 μm pore size membrane filter (DISMIC-25CS; Advantec Toyo Kaisha, Ltd., Japan). Then, 40 μL of the filtered sample was dried at 40˚C for 90 min using a vacuum constant temperature dryer (VOS-201SD, Eyela, Japan). After adding 20 μL of mixing solution (ethanol: DW: triethylamine = 2:2:1), the tube was mixed for 20 min using a Micro Mixer E-36 (TAITEC Corporation, Japan). The sample was heated at 40˚C for 60 min in a vacuum to dry. After adding 20 μL of the mixed solution (ethanol: DW: triethylamine: phenylisothiocyanate = 7:1:1:1), the tube was vortexed for 20 min. After vacuum drying at 40˚C for 90 min, the sample was kept at -30˚C until the amino acid analysis.

Detection of free amino acids in yolk was performed by HPLC (LC-2010CHT; Shimadzu Co. Ltd., Japan) using 4.6 μm columns (TSKgel ODS80Ts; 150 mm and 250 mm, Tosoh Corporation, Japan) at a flow rate of 1.0 mL/min. Gradient release was used with mobile phase A (60 mM acetic acid buffer solution: acetonitrile = 94:6) and mobile phase B (60 mM acetic acid buffer solution: acetonitrile = 60:40). Wavelengths of 254 nm were detected by a UV detector. Amino acid standards (B-type and H-type), L-asparagine, and L-glutamine were prepared in the same manner as the sample pretreatment. The concentration of free amino acid in yolk was determined from the peaks of the sample and standards [17].

## Free amino acid analysis of albumen

The frozen albumen sample was thawed and mixed thoroughly. Incubation was carried out at room temperature for 3 min to separate parts of the foam and liquid. The 500 μL of albumen sample, 500 μL of MilliQ water, and 1,000 μL of acetonitrile were mixed (albumen sample:

DW: acetonitrile = 1:1:2) and vortexed for 1 min. Then, centrifugation was performed at 22,000 × g for 5 min, and the lower layer was filtered with a 0.2 μm pore-sized filter.

The free amino acids of the albumen sample were analyzed by UHPLC (NexeraX2, Shimadzu 353 Co., Ltd., Japan) using Kinetex 2.6 μm (EVO C18 100 × 3.0 mm) for the column [17]. Gradient release was used in this study with mobile phase A (17 mM potassium dihydrogen phosphate and 3 mM dipotassium hydrogen phosphate) and mobile phase B (DW: acetonitrile: methanol = 15:45:40). For pre-column derivatization, 45 μL of mercaptopropionic acid, 22 μL of ortho-phthalaldehyde, and 7.5 μL of albumen sample were mixed and incubated at room temperature for 1 min. Then, 5 μL of fluorenylmethyl chloroformate was mixed with the solution and incubated at room temperature for 2 min. The 1 μL of the derivatized solution was injected into the column at a flow rate of 0.85 mL/min. A fluorescence detector RF-20AXS was used for wavelength detection. In this study, wavelengths of Ch1 EX 350 nm EM 450 nm and Ch2 EX 266 nm EM 305 nm were used [17].

### Statistical analysis

A one-way analysis of variance (ANOVA) with a significant threshold ($P < 0.05$) was performed to test the effect on genotype in egg traits and amino acids traits. Tukey's HSD test with a significant threshold ($P < 0.05$) was performed to see the difference in each pair of genotypes. Data were presented as mean ± standard deviation.

Pearson's correlation was estimated among all traits using the 'corrplot' package in R [29]. A significant threshold ($P < 0.05$) was used to express whether correlation coefficients equal zero. All statistical analyses were performed using R.

## Results

### Egg traits

Egg traits in five genotypes (ARA, KRH, UKO, NGY, and BOR) are indicated in Tables 1 and S1. In 10 traits (EW, YW, AW, SW, LLE, LSE, ST, SCL, SCR, and SCY), significant effects of genotype were observed ($P < 0.05$). Egg weight, albumen weight, and eggshell weight of BOR were the highest among the five genotypes. On the other hand, yolk weight tended to be higher in ARA, KRH, and NGY than BOR. Egg weight, albumen weight, yolk weight, eggshell weight,

**Table 1. Mean and standard deviation of 10 egg traits in five genotypes of chickens.**

| Trait[1] | Araucana Cross | Kurohisui | Ukokkei | Nagoya | Boris Brown | One-way ANOVA |
|---|---|---|---|---|---|---|
| | (ARA; $n = 10$) | (KRH; $n = 12$) | (UKO; $n = 12$) | (NGY; $n = 12$) | (BOR; $n = 12$) | P value |
| EW (g) | 56.7±3.0[a,b] | 54.3±5.2[b] | 39.4±2.6[c] | 56.7±4.3[b] | 61.6±3.6[a] | <0.001 |
| LLE (mm) | 55.3±1.6[b] | 55.8±2.8[a,b] | 49.3±1.6[c] | 55.5±1.8[b] | 58.1±2.1[a] | <0.001 |
| LSE (mm) | 43.0±1.3[a,b] | 41.7±1.2[b] | 37.8±0.9[c] | 42.7±1.2[a,b] | 43.7±0.9[a] | <0.001 |
| YW (g) | 16.7±1.8[a] | 17.2±1.8[a] | 12.5±0.6[b] | 17.5±1.2[a] | 16.6±1.0[a] | <0.001 |
| SW (g) | 6.9±0.6[a,b] | 6.9±1.0[a,b] | 4.8±0.4[c] | 6.4±0.8[b] | 7.6±0.5[a] | <0.001 |
| AW (g) | 33.1±3.4[b] | 30.3±3.4[b] | 22.1±1.9[c] | 32.8±3.1[b] | 37.4±2.8[a] | <0.001 |
| ST (mm) | 0.44±0.04[a] | 0.42±0.04[a,b] | 0.38±0.03[b] | 0.39±0.04[b] | 0.42±0.04[a,b] | 0.003 |
| SCL | 74.6±4.6[a] | 64.1±2.8[b] | 73.5±3.2[a] | 72.8±4.5[a] | 57.4±3.7[c] | <0.001 |
| SCR | -2.6±2.2[d] | 0.7±2.1[c] | 8.3±1.9[b] | 9.1±2.5[b] | 16.3±1.9[a] | <0.001 |
| SCY | 15.5±4.2[c] | 21.4±2.7[b] | 19.9±1.9[b] | 18.4±2.5[b,c] | 25.2±1.3[a] | <0.001 |

[1] Trait abbreviations are shown in Materials and Methods.

[a-d] The same superscript letter are not significantly different among the genotypes at $P > 0.05$ in each trait (Tukey's HSD test).

LLE, and LSE were lower in UKO than others. Eggshell redness was the highest in BOR (16.3 ± 1.9) and lowest in ARA (-2.6 ± 2.3).

## Yolk free amino acids

This study detected 19 yolk free amino acids: aspartic acid (Y_Asp), glutamic acid (Y_Glu), asparagine (Y_Asn), serine (Y_Ser), glutamine (Y_Gln), glycine (Y_Gly), histidine (Y_His), arginine (Y_Arg), threonine (Y_Thr), alanine (Y_Ala), proline (Y_Pro), tyrosine (Y_Tyr), valine (Y_Val), methionine (Y_Met), cysteine (Y_Cys), isoleucine (Y_Ile), Leucine (Y_Leu), phenylalanine (Y_Phe), and lysine (Y_Lys). Tables 2 and S1 show 19 yolk free amino acid contents among five genotypes. Significant effects on genotype (P < 0.05) were detected among 8 kinds of free amino acids (Y_Asp, Y_Glu, Y_Ser, Y_Gly, Y_Thr, Y_Tyr, Y_Cys, and Y_Leu). Yolk aspartic acid in NGY (55.7 ± 13.9 μg/mL), which is the highest concentration, was approximately twice as much as the lowest ARA (27.6 ± 7.0 μg/mL). Yolk glycine in NGY, BOR, and ARA were higher than the others. Yolk serine in NGY and UKO were the highest and lowest, respectively. On the other hand, yolk cysteine in NGY (1.8 μg/mL) was lower than the other genotypes. NGY and KRH had the lowest and highest content of yolk leucine, respectively. ARA and NGY had greater contents of yolk tyrosine, while UKO was the lowest. KRH and ARA had the highest and lowest content of yolk glutamic acid, respectively. KRH had remarkably higher yolk threonine (71.3 μg/mL) than the other genotypes.

**Table 2. Mean and standard deviation of 19 yolk amino acid traits in five genotypes of chickens.**

| Trait[1] | Araucana Cross | Kurohisui | Ukokkei | Nagoya | Boris Brown | One-way ANOVA |
|---|---|---|---|---|---|---|
| | (ARA; *n* = 10) | (KRH; *n* = 12) | (UKO; *n* = 12) | (NGY; *n* = 12) | (BOR; *n* = 12) | P value |
| Y_Asp | 27.6±7.0[c] | 42.9±12.7[a,b] | 32.1±8.9[b,c] | 55.7±13.9[a] | 33.8±9.3[b,c] | <0.001 |
| Y_Glu | 166.6±19.4[b] | 206.9±27.5[a] | 181.5±20.7[a,b] | 183.1±21.6[a,b] | 173.8±18.7[b] | 0.007 |
| Y_Asn | 35.0±4.5 | 34.8±4.5 | 33.3±4.0 | 34.5±3.4 | 35.6±3.1 | 0.740[ns] |
| Y_Ser | 57.9±6.1[a,b] | 59.9±5.6[a,b] | 54.2±4.9[b] | 61.2±5.0[a] | 59.7±4.9[a,b] | 0.037 |
| Y_Gln | 66.5±4.5 | 73.1±6.5 | 67.1±5.5 | 66.2±5.9 | 69.2±5.0 | 0.121[ns] |
| Y_Gly | 22.2±2.4 | 19.6±2.5 | 19.9±1.8 | 22.0±2.1 | 22.0±1.9 | 0.007 |
| Y_His | 17.6±3.1 | 16.6±2.9 | 21.3±5.5 | 18.3±2.2 | 18.8±3.4 | 0.051[ns] |
| Y_Arg | 75.4±10.5 | 78.0±8.4 | 76.2±10.7 | 76.1±5.4 | 77.4±7.9 | 0.977[ns] |
| Y_Thr | 61.0±6.6[b] | 71.3±9.8[a] | 61.6±5.3[b] | 60.0±4.7[b] | 60.0±3.7[b] | 0.002 |
| Y_Ala | 37.9±4.2 | 43.9±4.3 | 40.2±4.1 | 40.9±3.5 | 39.8±3.4 | 0.064[ns] |
| Y_Pro | 41.3±3.4 | 40.2±2.9 | 39.1±2.4 | 39.3±2.9 | 41.0±2.6 | 0.315[ns] |
| Y_Tyr | 75.2±7.6[a] | 70.3±6.1[a,b] | 67.1±6.4[b] | 74.3±5.4[a,b] | 69.8±5.4[a,b] | 0.020 |
| Y_Val | 68.7±7.4 | 68.3±6.3 | 62.8±5.8 | 63.4±4.8 | 66.3±4.4 | 0.103[ns] |
| Y_Met | 28.7±3.1 | 29.9±3.5 | 27.3±3.2 | 30.3±3.0 | 30.3±2.5 | 0.164[ns] |
| Y_Cys | 2.8±0.4[a] | 2.7±0.5[a] | 2.5±0.9[a,b] | 1.8±0.6[b] | 2.4±0.5[a,b] | 0.002 |
| Y_Ile | 64.1±6.4 | 62.7±6.3 | 59.2±5.8 | 63.6±4.4 | 60.8±4.1 | 0.248[ns] |
| Y_Leu | 120.9±13.1 | 125.3±12.7 | 113.4±12.4 | 112.2±8.7 | 122.6±9.4 | 0.050 |
| Y_Phe | 55.9±7.6 | 55.2±6.8 | 53.2±7.2 | 53.3±5.3 | 54.1±5.6 | 0.892[ns] |
| Y_Lys | 92.0±13.2 | 92.9±10.4 | 92.2±11.7 | 96.5±8.5 | 89.3±8.8 | 0.639[ns] |

[1] Trait abbreviations are shown in Materials and Methods. Amino acids concentrations are shown in μg/mL.

[ns] not significant (P > 0.05).

[a-d] The same superscript letter are not significantly different among the genotypes at P > 0.05 in each trait (Tukey';s HSD test).

**Table 3. Mean and standard deviation of 19 albumen amino acid traits in five genotypes of chickens.**

| Trait[1] | Araucana Cross (ARA; $n$ = 10) | Kurohisui (KRH; $n$ = 12) | Ukokkei (UKO; $n$ = 12) | Nagoya (NGY; $n$ = 12) | Boris Brown (BOR; $n$ = 12) | One-way ANOVA P value |
|---|---|---|---|---|---|---|
| A_Asp | 0.57±0.28[b] | 0.54±0.19[b] | 0.57±0.16[b] | 0.93±0.49[a] | 0.52±0.10[b] | 0.008 |
| A_Glu | 1.00±0.38[a,b] | 0.87±0.27[b] | 1.06±0.28[a,b] | 1.44±0.74[a] | 0.82±0.12[b] | 0.013 |
| A_Asn | 0.03±0.01[b] | 0.02±0.01[b] | 0.04±0.02[a,b] | 0.09±0.11[a] | 0.02±0.01[b] | 0.009 |
| A_Ser | 0.57±0.28[a,b] | 0.49±0.21[b] | 0.57±0.17[a,b] | 0.85±0.53[a] | 0.43±0.10[b] | 0.026 |
| A_Gln | 1.18±0.51[a,b] | 1.07±0.55[a,b] | 1.15±0.36[a,b] | 1.97±1.50[a] | 0.84±0.28[b] | 0.024 |
| A_His | 0.18±0.07[a,b] | 0.13±0.07[b] | 0.23±0.06[a] | 0.23±0.13[a] | 0.16±0.04[a,b] | 0.032 |
| A_Gly | 0.12±0.06 | 0.10±0.05 | 0.11±0.03 | 0.15±0.08 | 0.10±0.04 | 0.197[ns] |
| A_Thr | 0.52±0.27[a,b] | 0.45±0.19[a,b] | 0.51±0.18[a,b] | 0.69±0.41[a] | 0.38±0.10[b] | 0.088[ns] |
| A_Arg | 0.70±0.29 | 0.60±0.19 | 0.69±0.21 | 0.87±0.43 | 0.59±0.11 | 0.146[ns] |
| A_Ala | 0.34±0.19[a,b] | 0.31±0.14[a,b] | 0.39±0.13[a,b] | 0.50±0.31[a] | 0.24±0.07[b] | 0.042 |
| A_Tyr | 0.75±0.26[a,b] | 0.77±0.15[a,b] | 0.89±0.17[a,b] | 1.05±0.45[a] | 0.64±0.13[b] | 0.010 |
| A_Val | 0.61±0.30[a,b] | 0.51±0.22[a,b] | 0.66±0.16[a,b] | 0.76±0.41[a] | 0.45±0.10[b] | 0.072[ns] |
| A_Met | 0.49±0.23 | 0.38±0.17 | 0.54±0.10 | 0.58±0.30 | 0.37±0.10 | 0.067[ns] |
| A_Trp | 0.28±0.08[b] | 0.32±0.04[a] | 0.35±0.04[a] | 0.35±0.06[a] | 0.24±0.05[b] | <0.001 |
| A_Phe | 1.22±0.20[b] | 1.25±0.21[b] | 1.53±0.16[a] | 1.31±0.22[b] | 1.24±0.10[b] | 0.002 |
| A_Ile | 0.58±0.21 | 0.51±0.15 | 0.68±0.08 | 0.68±0.25 | 0.49±0.08 | 0.031 |
| A_Leu | 1.39±0.38 | 1.16±0.31 | 1.50±0.15 | 1.47±0.50 | 1.19±0.17 | 0.103[ns] |
| A_Lys | 0.03±0.02 | 0.03±0.02 | 0.04±0.02 | 0.05±0.02 | 0.04±0.04 | 0.633[ns] |
| A_Pro | 0.64±0.11 | 0.66±0.10 | 0.65±0.08 | 0.62±0.13 | 0.68±0.18 | 0.892[ns] |

[1] Trait abbreviations are shown in Materials and Methods. Amino acids concentrations are shown in μg/mL.

[ns] not significant (P > 0.05).

[a-d] The same superscript letter are not significantly different among the genotypes at P > 0.05 in each trait (Tukey';s HSD test).

## Albumen free amino acids

This study detected 19 albumen free amino acids: aspartic acid (A_Asp), glutamic acid (A_Glu), asparagine (A_Asn), serine (A_Ser), glutamine (A_Gln), histidine (A_His), glycine (A_Gly), threonine (A_Thr), arginine (A_Arg), alanine (A_Ala), tyrosine (A_Tyr), valine (A_Val), methionine (A_Met), tryptophan (A_Trp), phenylalanine (A_Phe), isoleucine (A_Ile), leucine (A_Ieu), lysine (A_Lys), and proline (A_Pro). Tables 3 and S1 indicate the results of 19 albumen free amino acid contents among five chicken genotypes. Significant effects of genotype (P < 0.05) were observed on 11 albumen amino acids (A_Asp, A_Glu, A_Asn, A_Ser, A_Gln, A_His, A_Ala, A_Tyr, A_Trp, A_Phe, and A_Ile). Among them, NGY showed the highest values in 7 albumen amino acids (A_Asp, A_Glu, A_Asn, A_Ser, A_Gln, A_Ala, and A_Tyr). Three albumen amino acids (A_His, A_Trp, and A_Ile) in NGY and UKO were the highest contents. Albumen phenylalanine in UKO (1.53 ± 0.16 μg/mL) was the highest. On the other hand, BOR had the lowest 9 albumen amino acids (A_Asp, A_Glu, A_Asn, A_Ser, A_Gln, A_Ala, A_Tyr, A_Trp, and A_Ile). KRH and ARA had the lowest contents of albumen histidine and phenylalanine, respectively.

## Phenotypic correlation among all egg traits

Pair-wise correlation coefficients among 48 traits were estimated using all 58 eggs (Fig 1). Across 10 egg traits, there were moderate to strong positive correlations among sizes and weights (EW, LLE, LSE, YW, SW, AW, and ST). Across 19 yolk amino acid traits, there were moderate to high positive correlation coefficients among each other, except for yolk cysteine.

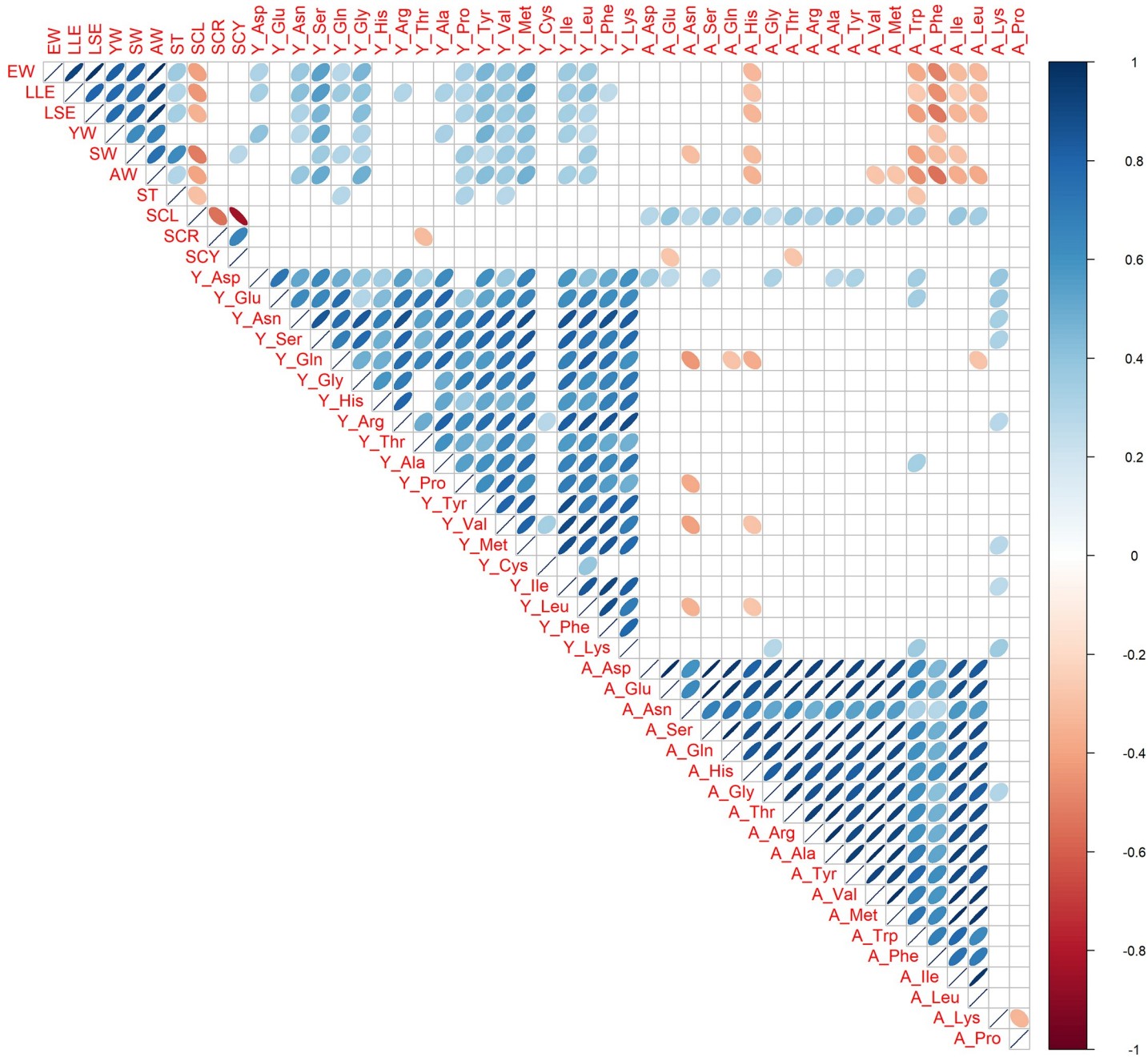

**Fig 1. Phenotypic correlations among all 48 egg traits including yolk and albumen free amino acids.** Ten egg traits, 19 yolk amino acids traits, and 19 albumen amino acids traits from a total of 58 hens (ARA, KRH, UKO, NGY, and BOR) were used. Trait abbreviations are shown in Materials and Methods and Results. Pearson's correlations are expressed by ellipses. Blue and red ellipses indicate positive and negative correlations in each pair, respectively (P < 0.05). Shapes of ellipses are depended on weak to strong correlations from almost circle to very sharp ellipses, respectively. Blank cells show no significance.

Across 19 albumen amino acid traits, moderate to strong positive phenotypic correlations were estimated, although there were no correlations between two traits (albumen lysine and albumen proline) and the others. Regardless of positive correlations estimated among traits within each trait category (size and weight traits, yolk amino acid traits, and albumen amino acid traits), there were no or weak phenotypic correlations among the trait categories.

Interestingly, there were nearly no correlations between yolk and albumen amino acid traits even in the same metabolites implying that yolk and albumen free amino acids are regulated by partially different genetic bases.

Phenotypic correlation coefficients among 48 traits were estimated in each genotype (S1–S5 Figs). A similar tendency to phenotypic correlations in all five genotypes (Fig 1) was basically seen in KRH, UKO, NGY, and BOR (S2–S5 Figs). In ARA (S1 Fig), there were uniquely moderate positive correlations between yolk and albumen amino acid traits.

## Discussion

In this study, we aimed to investigate the effects of genotype on egg traits, yolk free amino acid traits, and albumen free amino acid traits, using five chicken genotypes (ARA, KRH, UKO, NGY, and BOR). We observed significant effects of genotype on 10 egg traits, 8 yolk amino acids, and 11 albumen amino acids contents. Moderate to strong positive phenotypic correlations were seen among traits within each trait category (size and weight traits, yolk amino acid traits, and albumen amino acid traits), whereas there were nearly no correlations between yolk and albumen amino acid traits even in the same metabolites except for ARA. Thus, these results suggest that many egg traits including yolk and albumen free amino acids can be modified by genetic background.

Egg weights in UKO (39.4 ± 2.6 g) and NGY (56.7 ± 4.3 g) in this study were similar to those in UKO (38.5 ± 2.6 g) and NGY (58.3 g) reported by Koketsu and Toyosaki [30] and Nakamura et al. [31]. BOR showed the highest egg weight (61.6 ± 3.6 g) in the five chicken genotypes. However, Kamisoyama et al. [32] and Shimmura et al. [33] reported that egg weights of BOR at 30 weeks of age and 80 weeks of age in the conventional cage environment were 65.1 ± 1.3 g and 66.7 ± 3.1 g, respectively. From these results, the egg weight of BOR used in this study will be slightly smaller than the general eggs from BOR. In the Tago Poultry Farm, all the birds were kept in a floor-rearing environment, which is very far from the conventional cage-laying system. Therefore, the environmental difference should be tested in the future.

Of 10 egg traits, the most variety was seen in eggshell redness among five chicken genotypes. Higher eggshell redness values (more than 0) indicate a higher degree of redness, while lower values (less than 0) indicate a higher degree of greenness [34]. From the previous studies, brown eggs laid by RIR were approximately 11.2 to 14.2 in eggshell redness [24], whereas white eggs laid by White Leghorn were approximately -0.3 to 0.2 [35]. In this study, BOR showed 16.3 ± 1.9 in eggshell redness indicating that the eggshell redness of BOR was quite high. ARA showed -2.6 ± 2.2 in eggshell redness, which indicated a higher degree of greenness than the other chicken genotypes. On the other hand, the eggshell yellowness of ARA was 15.5 ± 4.2 in this study. Since higher and lower eggshell yellowness values (more and less than 0) indicate a higher degree of yellowness and blueness, respectively [34], the present ARA produced yellow eggshell color rather than blue. In general, it is well known that the pure breed of Araucana produces blue eggs [36–38]. Since the ARA egg in this study was close to green rather than blue, the ARA must be crossbred between purebred Araucana and brown layers. Given that there is some genetic contribution of the brown layer to the present ARA, green-colored eggshell will be reasonable.

Yolk and albumen free amino acids among five chicken genotypes were compared to check effects on the genetic differences. This study was conducted under completely the same feeding environment for all hens. Rearing and feeding environments were fixed in the floor rearing system and mixed feed, and moreover, age at egg sampling was also fixed as 72 weeks of age. Therefore, major parts of the phenotypic variance in this study must be caused by differences

in genetic background. This study revealed significant genetic effects in 8 yolk amino acids and 11 albumen amino acids. Of them, the ranking of free amino acid contents was seen as KRH > NGY > ARA > BOR > UKO in yolk and NGY > UKO > ARA, KRH > BOR in albumen. In both yolk and albumen, NGY and BOR showed higher and lower contents, respectively. Our previous study found a significant impact on the genetic difference in 20 yolk free amino acid traits (Y_Asp, Y_Glu, Y_Asn, Y_Ser, Y_Gln, Y_Gly, Y_His, Y_Arg, Y_Thr, Y_Ala, Y_Pro, Y_GABA, Y_Tyr, Y_Val, Y_Met, Y_Cys, Y_Ile, Y_Leu, Y_Phe, and Y_Lys) and 15 albumen free amino acid traits (A_Asp, A_Glu, A_Ser, A_Gly, A_Thr, A_Arg, A_Ala, A_Tyr, A_Val, A_Met, A_Trp, A_Phe, A_Ile, A_Leu, and A_Pro) using five breeds (AUS, NGY, RIR, SHA, and UKO) and two $F_1$ hybrids (NGYxRIR and SHAxRIR) of chickens under cage rearing system [17]. This study revealed significant effects of genetic difference on 8 yolk free amino acids (Y_Asp, Y_Glu, Y_Ser, Y_Gly, Y_Thr, Y_Tyr, Y_Cys, and Y_Leu) and 11 albumen free amino acids (A_Asp, A_Glu, A_Asn, A_Ser, A_Gln, A_His, A_Ala, A_Tyr, A_Trp, A_Phe, and A_Ile) using five chicken genotypes (ARA, KRH, UKO, NGY, and BOR) under floor rearing system. In comparison with these two experiments, 8 yolk amino acids (Y_Asp, Y_Glu, Y_Ser, Y_Gly, Y_Thr, Y_Tyr, Y_Cys, and Y_Leu) and 8 albumen amino acids (A_Asp, A_Glu, A_Ser, A_Ala, A_Tyr, A_Trp, A_Phe, and A_Ile) were commonly altered by genetic factor suggesting that these free amino acids may have relatively higher heritability estimates than the others.

Essential amino acids for humans are His, Arg, Thr, Tyr, Val, Met, Ile, Leu, Phe, and Lys [39]. Of them, 3 yolk amino acids (Thr, Tyr, and Leu) and 4 albumen amino acids (His, Tyr, Phe, and Ile) were significantly different among five genotypes in this study. KRH showed potential to increase these essential amino acids in the yolk, whereas NGY and UKO tended to be higher in albumen. Using these genotypes, essential amino acid-enriched eggs may be created. However, this study analyzed free amino acids only. Therefore, the constituent amino acids should be analyzed for accumulating effect of genotype on nutritional values of yolk and albumen. In addition, free amino acids are related to food taste [26,40]. Zhao et al. [41] and Iwaniak et al. [42] have summarized taste-active amino acids, which are umami taste (Glu, Asp, Ala, and Tyr), sweet taste (Met, Ala, Gly, Pro, Ser, Val, and Lys), sour taste (Asp, Glu, and Lys), bitter taste (Pro, Gly, Val, Leu, Tyr, Phe, His, Lys, Ile, Arg, and Trp), salty taste (Asp), and astringent taste (Lys). This study found significantly altered 6 yolk amino acids (Y_Asp, Y_Glu, Y_Ser, Y_Gly, Y_Tyr, and Y_Leu) and 9 albumen amino acids (A_Asp, A_Glu, A_Ser, A_His, A_Ala, A_Tyr, A_Trp, A_Phe, and A_Ile) indicating that taste added designer eggs will be created by the genetic difference in the future egg industry. Although amino acid and taste sensor analyses have been conducted with yolk and albumen [25], it is still unclear what degree of difference in free amino acid contents actually affect taste difference in complex food materials such as yolk and albumen. Thus, accumulating knowledge of the relationship between taste and amino acids in eggs is needed using human sensory evaluation and instrumental analysis.

Using varieties of chicken genotypes, significant effects on genetic differences were found in yolk and albumen free amino acid contents. Until now, genome-wide association study (GWAS) and quantitative trait locus (QTL) mappings have been performed for egg traits such as egg weight, eggshell color, yolk size, and albumen height for identifying genetic variants [10–14]. However, there are few reports of GWAS and QTL studies for egg yolk and albumen amino acid traits so far [17]. This study could contribute to accumulating evidence that genetic factors are important to regulate egg amino acid contents with previous works [15–17]. Given that yolk and albumen free amino acid contents are general quantitative traits as well as egg weight, several genetic loci can be found by genomics studies with a large segregating

population using indigenous chicken breeds [13]. We plan to perform these genomics studies for the understanding complex genetic architecture of egg amino acid contents in chickens.

It was found that moderate to high phenotypic correlations among free amino acid traits within each yolk and albumen, but no or weak correlations of amino acid traits between yolk and albumen. The relationship of phenotypic correlation is basically supported by Goto et al. [17]. However, this study revealed a unique feature in ARA, which shows moderate positive correlations between yolk and albumen amino acids. This may imply that free amino acid balance in yolk and albumen can be regulated by selecting chicken genotypes. Since egg components have antioxidant properties, antimicrobial activities, immunomodulatory, anticancer, and antihypertensive activities [43,44], phenotypic correlations among many egg components should be investigated to modify balance in bioactive compounds of eggs in the future.

In summary, the genetic factor could affect not only the size and weight of the egg and eggshell color but also yolk and albumen free amino acids. Phenotypic correlation analyses revealed unique positive correlations of amino acids between yolk and albumen in ARA. Since there are some possibilities that amino acid performance can be altered by genotypes by several rearing systems, further investigation of genotype by environment interaction should be conducted.

## Supporting information

**S1 Fig. Phenotypic correlations among all 48 egg traits in ARA.** Ten egg traits, 19 yolk amino acids traits, and 19 albumen amino acids traits from 10 hens (ARA) were used. Trait abbreviations are shown in Materials and Methods and Results. Pearson's correlations are expressed by ellipses. Blue and red ellipses indicate positive and negative correlations in each pair (P < 0.05). Blank cells show no significance.
(TIFF)

**S2 Fig. Phenotypic correlations among all 48 egg traits in KRH.** Ten egg traits, 19 yolk amino acids traits, and 19 albumen amino acids traits from 12 hens (KRH) were used. Trait abbreviations are shown in Materials and Methods and Results. Pearson's correlations are expressed by ellipses. Blue and red ellipses indicate positive and negative correlations in each pair (P < 0.05). Blank cells show no significance.
(TIFF)

**S3 Fig. Phenotypic correlations among all 48 egg traits in UKO.** Ten egg traits, 19 yolk amino acids traits, and 19 albumen amino acids traits from 12 hens (UKO) were used. Trait abbreviations are shown in Materials and Methods and Results. Pearson's correlations are expressed by ellipses. Blue and red ellipses indicate positive and negative correlations in each pair (P < 0.05). Blank cells show no significance.
(TIFF)

**S4 Fig. Phenotypic correlations among all 48 egg traits in NGY.** Ten egg traits, 19 yolk amino acids traits, and 19 albumen amino acids traits from 12 hens (NGY) were used. Trait abbreviations are shown in Materials and Methods and Results. Pearson's correlations are expressed by ellipses. Blue and red ellipses indicate positive and negative correlations in each pair (P < 0.05). Blank cells show no significance.
(TIFF)

**S5 Fig. Phenotypic correlations among all 48 egg traits in BOR.** Ten egg traits, 19 yolk amino acids traits, and 19 albumen amino acids traits from 12 hens (BOR) were used. Trait abbreviations are shown in Materials and Methods and Results. Pearson's correlations are

expressed by ellipses. Blue and red ellipses indicate positive and negative correlations in each pair (P < 0.05). Blank cells show no significance.
(TIFF)

**S1 Table. Degree of freedom, F value, and P value of one-way ANOVA.**
(DOCX)

**S1 File.**
(XLSX)

## Acknowledgments

We are grateful to Drs. Masaaki Hanada and Masafumi Tetsuka for lending laboratory equipment. We also appreciate David Campbell for his valuable advice on English proofreading. We thank all members of the Animal Breeding and Genetics Research Group at the Obihiro University of Agriculture and Veterinary Medicine for their continuous supports.

## Author Contributions

**Conceptualization:** Tatsuhiko Goto.

**Formal analysis:** Kenji Nishimura, Tatsuhiko Goto.

**Funding acquisition:** Kenji Nishimura, Tatsuhiko Goto.

**Investigation:** Kenji Nishimura, Daichi Ijiri, Saki Shimamoto, Masahiro Takaya, Akira Ohtsuka, Tatsuhiko Goto.

**Project administration:** Tatsuhiko Goto.

**Supervision:** Tatsuhiko Goto.

**Writing – original draft:** Kenji Nishimura, Tatsuhiko Goto.

**Writing – review & editing:** Kenji Nishimura, Daichi Ijiri, Saki Shimamoto, Masahiro Takaya, Akira Ohtsuka, Tatsuhiko Goto.

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
