## [Decision Letter · Decision Letter 0]

15 Jun 2021

PONE-D-21-14031

Genetic effect on free amino acid contents of egg yolk and albumen using five different chicken genotypes under floor rearing system

PLOS ONE

Dear Dr. Tatsuhiko Goto,

Thank you for submitting your manuscript to PLOS ONE. After careful consideration, we feel that it has merit but does not fully meet PLOS ONE’s publication criteria as it currently stands. Therefore, we invite you to submit a revised version of the manuscript that addresses the points raised during the review process.

We look forward to receiving your revised manuscript.

Kind regards,

Ewa Tomaszewska, DVM Ph.D

Academic Editor

PLOS ONE

Journal Requirements:

Reviewers' comments:

Reviewer's Responses to Questions

**Comments to the Author**

1. Is the manuscript technically sound, and do the data support the conclusions?

Reviewer #1: Partly

Reviewer #2: Partly

2. Has the statistical analysis been performed appropriately and rigorously? 

Reviewer #1: Yes

Reviewer #2: Yes

3. Have the authors made all data underlying the findings in their manuscript fully available?

Reviewer #1: Yes

Reviewer #2: Yes

4. Is the manuscript presented in an intelligible fashion and written in standard English?

Reviewer #1: Yes

Reviewer #2: No

5. Review Comments to the Author

Reviewer #1: The paper entitled "Genetic effect on free amino acid contents of egg yolk and albumen using five different chicken genotypes under floor rearing system" presents interesting results, but requires major corrections.

Abstract

In my opinion, including the letter abbreviations of each hen’s breed is not necessary in the abstract, especially they are given as full in the text of the manuscript

Line 40 Please replace "color of the egg" with "color of the eggshell"

Introduction

Lines 62-63- I suggest replacing "number of eggs" with "laying performance" for clarity

Please change "chicken" to "hens" or laying hens when referring to typical laying hens such as in Line 63

Line 72-73 It is worth to give an example of research concerning more than two breeds extremely different in eggshell colour, as paper:

• Drabik, K., Karwowska, M., Wengerska, K., Próchniak, T., Adamczuk, A., & Batkowska, J. (2021). The variability of quality traits of table eggs and eggshell mineral composition depending on hens' breed and eggshell color. Animals, 11(5), 1204.

• Goger, H., Demirtas, S. E., & Yurtogullari, S. (2016). A selection study for improving eggshell colour in two parent lines of laying hens and their hybrids. Italian Journal of Animal Science, 15(3), 390-395.

Line 81 Please check the sentence about glutamic acid carefully, it is not understandable

Line 94 According to most classifications of housing systems we are talking about free range or floor systems. Please clarify.

Material and methods

Line 101- please explain "Araucana cross", it was not a pure breed? If not please indicate clearly the genetic origin of the hybrid . If it was not a pure breed it is difficult to state clearly the genotype effect.

The material for the study consisted of 58 eggs obtained from 58 hens. So one egg per laying hen was analysed? Please explain.

The sample size raises concerns. The small number of eggs subjected to analysis may cause problems in correct analysis of the results obtained, mostly due to the laying hens are characterized by considerable variability not only individual but also within the subsequent laying cycles.

Line 103-104- Please indicate clearly the rearing system (as in line 94) or indicate what was the characteristic of the system used in the study. In addition, it is useful to indicate at least the basic rearing parameters such as lighting programme, feeding (restricted or ad libitum) etc.

Line 107-109 - The description of the feeding is insufficient. Please give the feed composition including the level of limiting amino acids.

Line 121- There are several ways and scales to measure the shell color. Have you used the one proposed by CIE i.e. L*a*b*? Please include this information in the material and methods section.

You describe in detail the description of the methodology for the determination of amino acids by UHPLC, but the description of the HPLC method is very limited. Please standardize by stating the basic analysis parameters (column, phase system etc.)

Table 1 is unreadable. If possible please divide it into two parts i.e. egg quality characteristics and amino acid compositions of egg albumen and yolk. Additionally, there is no need to give df or F-test values when presenting exact P-value

Why does in Table S1 the Tukey test was used instead of the typical t-test when two groups are compared ?

Please add titles and legends to the tables attached as supplemental material. In accordance with editorial requirements, titles are recommended. In my opinion, they will make it much easier for readers to understand the contents of the tables.

In terms of Pearson correlations, does the size of the ellipse used matter? While its color is clearly explained in the text and with the attached scale, no such information is available for its size. Please clarify in the manuscript when first describing the correlation.

The discussion variation in terms of individual components of shell color (redness, greeness) is indicated while result showed in Table 1 suggests the use of a different method of measurement. Clarification is needed in both the Materials and Methods section and in the Discussion.

Line 20 (Discussion) Please explain why there is so much variation within color in Araucana. The data indicate a significant variation in colour intensity which is quite rare in pure breeds.

The discussion needs the improvement. At present, only fragments have an explanatory function for the results described by the authors. The vast majority of it especially within the free amino acids focuses on a comparison to previous studies by the authors. More emphasis should be placed on explaining phenomena than on comparing results.

Please check the References list carefully, also the formatting should be unified (i.e. dot after the initial of name).

Reviewer #2: General Comments:

1) The manuscript describes the effect of genotype on egg quality variables and free amino acid contents of yolk and albumen of eggs.

2) The study is well executed in terms of analysis and design. However, there are numerous grammatical mistakes in addition to poor language use. The authors should seek help from a native English-speaking colleague or a language editing service.

3) The authors have used the free-range floor system in their study. However, the rearing system was not used as factor in this study. Therefore, this should be removed throughout the study. Speculation on the part of environment or rearing system should be removed.

4) I am not entirely convinced of the originality or novelty of the study as the authors have compared different genotypes without any control group or a representative genotype as a benchmark. I was unable to see any sentence regarding the authenticity of the used genotypes (crossbred, purebred, etc.). Therefore, making a comparison is misleading at least in this manuscript. The authors should have described the genotypes/breeds properly in the materials and methods section of the manuscript. There should be a comparison with a representative breed as a benchmark (might be from white leghorn).

5) The authors have used abbreviations at their own liberty that should be avoided. Abbreviations should be defined first and used later on in the manuscript. Abstract should have separate abbreviations. Remaining manuscript should have abbreviations independent of the abstract.

Specific comments:

Specific comments and corrections (made by me) are given in the attached file.

6. PLOS authors have the option to publish the peer review history of their article (what does this mean?). If published, this will include your full peer review and any attached files.

Reviewer #1: No

Reviewer #2: No

---

## [Author Response · Author response to Decision Letter 0]

12 Jul 2021

PONE-D-21-14031

Genetic effect on free amino acid contents of egg yolk and albumen using five different chicken genotypes under floor rearing system

PLOS ONE

Dear Dr. Tatsuhiko Goto,

Thank you for submitting your manuscript to PLOS ONE. After careful consideration, we feel that it has merit but does not fully meet PLOS ONE’s publication criteria as it currently stands. Therefore, we invite you to submit a revised version of the manuscript that addresses the points raised during the review process.

[Authors’ response] Thank you very much for your contribution to evaluate our manuscript. We were careful to revise the manuscript according to the suggestions by two reviewers. We are glad if we are able to receive your re-evaluation.

We look forward to receiving your revised manuscript.

Kind regards,

Ewa Tomaszewska, DVM Ph.D

Academic Editor

PLOS ONE

Journal Requirements:

[Authors’ response] We confirmed. 

[Authors’ response] We confirmed. 

Reviewers' comments:

Reviewer's Responses to Questions

Comments to the Author

1. Is the manuscript technically sound, and do the data support the conclusions?

Reviewer #1: Partly

Reviewer #2: Partly

2. Has the statistical analysis been performed appropriately and rigorously? 

Reviewer #1: Yes

Reviewer #2: Yes

3. Have the authors made all data underlying the findings in their manuscript fully available?

Reviewer #1: Yes

Reviewer #2: Yes

4. Is the manuscript presented in an intelligible fashion and written in standard English?

Reviewer #1: Yes

Reviewer #2: No

5. Review Comments to the Author

Reviewer #1: The paper entitled "Genetic effect on free amino acid contents of egg yolk and albumen using five different chicken genotypes under floor rearing system" presents interesting results, but requires major corrections.

[Authors’ response] Thank you very much for your contribution to improve our manuscript. According to your advices, we have revised as much as possible. We are so happy if you re-evaluate our manuscript. 

Abstract

In my opinion, including the letter abbreviations of each hen’s breed is not necessary in the abstract, especially they are given as full in the text of the manuscript

[Authors’ response] We deleted the abbreviations of genotypes in Abstract (L30-31, L38). In the text of the manuscript, we would like to use the abbreviations. 

Line 40 Please replace "color of the egg" with "color of the eggshell"

[Authors’ response] We have revised (L40, L370-371).

Introduction

Lines 62-63- I suggest replacing "number of eggs" with "laying performance" for clarity

[Authors’ response] We have revised (L61). 

Please change "chicken" to "hens" or laying hens when referring to typical laying hens such as in Line 63

[Authors’ response] We have revised (L62).

Line 72-73 It is worth to give an example of research concerning more than two breeds extremely different in eggshell colour, as paper:

• Drabik, K., Karwowska, M., Wengerska, K., Próchniak, T., Adamczuk, A., & Batkowska, J. (2021). The variability of quality traits of table eggs and eggshell mineral composition depending on hens' breed and eggshell color. Animals, 11(5), 1204.

• Goger, H., Demirtas, S. E., & Yurtogullari, S. (2016). A selection study for improving eggshell colour in two parent lines of laying hens and their hybrids. Italian Journal of Animal Science, 15(3), 390-395.

[Authors’ response] Thank you very much. These papers look nice. We cited them (L68, L73, L429-431, L444-447). 

Line 81 Please check the sentence about glutamic acid carefully, it is not understandable

[Authors’ response] We revised (L80-81). 

Line 94 According to most classifications of housing systems we are talking about free range or floor systems. Please clarify.

[Authors’ response] We used floor rearing system. We revised it throughout the manuscript. 

Material and methods

Line 101- please explain "Araucana cross", it was not a pure breed? If not please indicate clearly the genetic origin of the hybrid . If it was not a pure breed it is difficult to state clearly the genotype effect.

[Authors’ response] Thank you very much. We used Araucana cross (ARA) in this study, which will be hybrids (not breed itself). Although we requested the origin of ARA to a breeding company, they wanted to be business secret what breeds/lines do they use. KRH will be hybrid (not breed), which is a part of business secret in the breeding company. 

Regarding the genotype effect, we think that there is no problem. Since a Brown layer (BOR) is created from a cross between RIR and White Plymouth Rock, BOR is hybrid commercial hen (not pure breed). BOR is one of the most famous Brown hybrid layer. Since the hybrid hens usually show the stable egg characteristics, we can evaluate the genotype effect fairly. In addition, genotype will be suitable term than breed, when we compare the performance among breeds and hybrids (Antova et al. Journal of the Science of Food and Agriculture 99 (13): 5890–5898. 2019; Fanatico et al. Poultry Science 84:1785–1790. 2005. Lewis et al. Meat Science 45 (4): 501-516, 1997; Castellini et al. Meat Science 60: 219–225. 2002.). 

We described which hens are pure breed and hybrid (L30-31, L101-103). 

The material for the study consisted of 58 eggs obtained from 58 hens. So one egg per laying hen was analysed? Please explain.

[Authors’ response] Eggs are collected every day in Tago Poultry Farm. Therefore, we collected all 58 eggs in a day, which mean that 58 different hens produced them. We analyzed one egg per hen. 

The sample size raises concerns. The small number of eggs subjected to analysis may cause problems in correct analysis of the results obtained, mostly due to the laying hens are characterized by considerable variability not only individual but also within the subsequent laying cycles.

[Authors’ response] Thank you very much for valuable comment. Actually, we agree with your suggestion. Ideally, we should collect and analyze more eggs to reduce probability of false positive. However, the present results look suitable using five genotypes and more than 10 eggs per genotype. 

Because the Tago Poultry Farm is small-scale under floor rearing system, each genotype is kept less than 50 hens. We wanted to collect eggs from more than 10 different hens in each genotype. 

Regarding laying cycle, we hypothesize that egg amino acid traits will be varied at early, middle, and late laying stages within breed. However, there is no evidence until now. Therefore, we will analyze the stage effect on egg amino acid traits in the future. 

Line 103-104- Please indicate clearly the rearing system (as in line 94) or indicate what was the characteristic of the system used in the study. In addition, it is useful to indicate at least the basic rearing parameters such as lighting programme, feeding (restricted or ad libitum) etc.

[Authors’ response] The hens were reared under the 16 h light and 8 h dark photoperiod cycle with free access to feeds and water. We added it (L104-105). 

Line 107-109 - The description of the feeding is insufficient. Please give the feed composition including the level of limiting amino acids.

[Authors’ response] The Tago Poultry Farm uses commercial diet formulated to satisfy the requirements of layers (16.5% CP, 2770 ME kcal/kg; Chubu Shiryo Co., Ltd., Japan) for all the genotypes. 

Line 121- There are several ways and scales to measure the shell color. Have you used the one proposed by CIE i.e. L*a*b*? Please include this information in the material and methods section.

[Authors’ response] We used eggshell color in units of the CIE L*a*b* color space. We added it (L120). 

You describe in detail the description of the methodology for the determination of amino acids by UHPLC, but the description of the HPLC method is very limited. Please standardize by stating the basic analysis parameters (column, phase system etc.)

[Authors’ response] We added the basic analysis parameters including column, phase, and detection (L143-147). 

Table 1 is unreadable. If possible please divide it into two parts i.e. egg quality characteristics and amino acid compositions of egg albumen and yolk. Additionally, there is no need to give df or F-test values when presenting exact P-value

[Authors’ response] Thank you very much. We divided the previous Table 1 into three Tables (Tables 1-3). Because df and F value are important statistical information, we added the detailed information in Table S1 rather than main Tables. 

Why does in Table S1 the Tukey test was used instead of the typical t-test when two groups are compared ?

[Authors’ response] Post-hoc test should mind the multiple comparison. Therefore, we usually use Tukey’s HSD test to keep alpha level is below 0.05. Typical t-test should be used for comparison of three or more groups with the significant threshold corrected by Bonferroni’s multiple comparison. Tukey’s HSD test is robust method with multiple comparison as post-hoc test following ANOVA. 

Please add titles and legends to the tables attached as supplemental material. In accordance with editorial requirements, titles are recommended. In my opinion, they will make it much easier for readers to understand the contents of the tables.

[Authors’ response] Thank you very much. We added titles of supplemental materials. Regarding phenotypic correlations in five genotypes, we think that supplementary figures will be better than the Tables, because matrix from 48 traits is large in each Table. 

In terms of Pearson correlations, does the size of the ellipse used matter? While its color is clearly explained in the text and with the attached scale, no such information is available for its size. Please clarify in the manuscript when first describing the correlation.

[Authors’ response] Shapes of ellipses are depended on weak to strong correlations from almost circle to very sharp ellipses, respectively. We added the description in Figure legend (L248-249). 

The discussion variation in terms of individual components of shell color (redness, greeness) is indicated while result showed in Table 1 suggests the use of a different method of measurement. Clarification is needed in both the Materials and Methods section and in the Discussion.

[Authors’ response] We added the information of CIE L*a*b* color space in Materials and Methods (L120). So, we think the discussion will be much clear. 

Line 20 (Discussion) Please explain why there is so much variation within color in Araucana. The data indicate a significant variation in colour intensity which is quite rare in pure breeds.

[Authors’ response] We used Araucana cross (ARA) in this study, which will be hybrids (not breed itself). We suppose that ARA is created from a cross between purebred Araucana and Brown layer. So, hybrid ARA hens will make the variation of eggshell color phenotype. 

The discussion needs the improvement. At present, only fragments have an explanatory function for the results described by the authors. The vast majority of it especially within the free amino acids focuses on a comparison to previous studies by the authors. More emphasis should be placed on explaining phenomena than on comparing results.

[Authors’ response] Regarding explain phenomena, we added the description about genetic architecture underlying variation in egg amino acid contents. We also added our future perspective in Discussion (L340-351). 

Please check the References list carefully, also the formatting should be unified (i.e. dot after the initial of name).

[Authors’ response] We found minor mistakes and revised them (L448). 

Reviewer #2: General Comments:

1) The manuscript describes the effect of genotype on egg quality variables and free amino acid contents of yolk and albumen of eggs.

[Authors’ response] Thank you very much for your contribution to improve our manuscript. According to your advices, we have revised as much as possible. We are so happy if you re-evaluate our manuscript. 

2) The study is well executed in terms of analysis and design. However, there are numerous grammatical mistakes in addition to poor language use. The authors should seek help from a native English-speaking colleague or a language editing service.

[Authors’ response] Thank you very much for your comment. Since we found the grammatical mistakes in parts thanks to your specific comments and corrections, we have revised throughout the manuscript. 

Although we usually receive language editing service before submitting, minor reviewers cannot accept it in many International Journals. Since major reviewers accept our English, we think it will be OK at minimum level of scientific paper. However, we are very happy if you let us know critical points concretely, because we are not native English speakers. 

3) The authors have used the free-range floor system in their study. However, the rearing system was not used as factor in this study. Therefore, this should be removed throughout the study. Speculation on the part of environment or rearing system should be removed. 

[Authors’ response] Thank you very much for your comment. We think the environmental information should be important to evaluate performance of egg production in future. Under the floor rearing system, hens can move freely and therefore their performance may change in comparison with cage system. Thus, we compare the egg weight of BOR under conventional cage system in the previous reports (Kamisoyama et al., 2010 and Shimmura et al., 2007) with the present results under floor rearing system. The discussion based on the comparison of scientific data using the previous reports will be effective. In addition, Goto et al [17] have reported egg amino acid traits using 7 genotypes under conventional cage system using common breeds (NGY and UKO). Therefore, we can discuss the differences between our present results of NGY and UKO under floor rearing system and the previous cage results. We think that those are informative indications (not just speculation) for future analyses. 

Moreover, we have several evidence that rearing environment (conventional cage vs floor rearing system) affect significant changes in the contents of egg amino acids (Goto et al., in preparation). Therefore, we would like to mention the description of rearing system in this study. 

4) I am not entirely convinced of the originality or novelty of the study as the authors have compared different genotypes without any control group or a representative genotype as a benchmark. I was unable to see any sentence regarding the authenticity of the used genotypes (crossbred, purebred, etc.). Therefore, making a comparison is misleading at least in this manuscript. The authors should have described the genotypes/breeds properly in the materials and methods section of the manuscript. There should be a comparison with a representative breed as a benchmark (might be from white leghorn).

[Authors’ response] Thank you very much. We added the detailed information about genotypes/breeds (L30-31, L101-103). 

Regarding the benchmark, we think BOR will be better because BOR is the world famous Brown layer (representative hybrid line). We previous reported the performance of classical type of White Leghorn (CB strain) (Goto et al., Animal Genetics 42: 634–641. 2011; Journal of Poultry Science 51: 118–129. 2014; Journal of Poultry Science 51: 375–386. 2014; Journal of Poultry Science 52; 81–87. 2015), which indicating very far from the performance of White Leghorn lines in breeding companies of the modern layer industry. In comparison with variable strains of White Leghorn breed, BOR is corresponding to recent performance of Brown layer in the world. Since the hybrid hens usually show the stable egg characteristics, we can evaluate the genotype effect fairly. Therefore, we think BOR will be suitable as a benchmark. 

Regarding originality and novelty of this study, we think that almost all data is firstly shown and therefore this study has high originality and novelty. In Japan, there are around 50 indigenous breeds. Some of them are used for egg and meat production. However, egg amino acid traits are not analyzed in most breeds. Novel finding of phenotypic difference using untapped breeds (not only Japanese but also the world’s) is very important and will be the starting point for deeper analysis. Since these genetic resources have a wide variety of genetic background, we can find new findings using these genetic resources in the future genomics studies (Goto and Tsudzuki, Journal of Poultry Science 54: 1–12. 2017). We added the descriptions (L340-351). 

5) The authors have used abbreviations at their own liberty that should be avoided. Abbreviations should be defined first and used later on in the manuscript. Abstract should have separate abbreviations. Remaining manuscript should have abbreviations independent of the abstract. 

[Authors’ response] We have revised. We deleted abbreviations in the Abstract and defined abbreviations firstly seen. 

Specific comments:

Specific comments and corrections (made by me) are given in the attached file.

[Authors’ response] We have never received the specific comments and corrections in the word file with track change. Almost all changes were approved. Thanks to your concrete advices, our manuscript can be improved thoroughly. We have revised throughout our manuscript. Thank you very much, again. 

6. PLOS authors have the option to publish the peer review history of their article (what does this mean?). If published, this will include your full peer review and any attached files.

Do you want your identity to be public for this peer review? For information about this choice, including consent withdrawal, please see our Privacy Policy.

Reviewer #1: No

Reviewer #2: No

---

## [Decision Letter · Decision Letter 1]

12 Aug 2021

PONE-D-21-14031R1

Genetic effect on free amino acid contents of egg yolk and albumen using five different chicken genotypes under floor rearing system

PLOS ONE

Dear Dr. Tatsuhiko Goto,

Thank you for submitting your manuscript to PLOS ONE. After careful consideration, we feel that it has merit but does not fully meet PLOS ONE’s publication criteria as it currently stands. Therefore, we invite you to submit a revised version of the manuscript that addresses the points raised during the review process.

We look forward to receiving your revised manuscript.

Kind regards,

Ewa Tomaszewska, DVM Ph.D

Academic Editor

PLOS ONE

Journal Requirements:

Reviewers' comments:

Reviewer's Responses to Questions

**Comments to the Author**

1. If the authors have adequately addressed your comments raised in a previous round of review and you feel that this manuscript is now acceptable for publication, you may indicate that here to bypass the “Comments to the Author” section, enter your conflict of interest statement in the “Confidential to Editor” section, and submit your "Accept" recommendation.

Reviewer #1: All comments have been addressed

Reviewer #2: (No Response)

2. Is the manuscript technically sound, and do the data support the conclusions?

Reviewer #1: Yes

Reviewer #2: Partly

3. Has the statistical analysis been performed appropriately and rigorously? 

Reviewer #1: Yes

Reviewer #2: Yes

4. Have the authors made all data underlying the findings in their manuscript fully available?

Reviewer #1: (No Response)

Reviewer #2: Yes

5. Is the manuscript presented in an intelligible fashion and written in standard English?

Reviewer #1: Yes

Reviewer #2: No

6. Review Comments to the Author

Reviewer #1: The authors made the suggested revisions to the manuscript. In addition, in situations of disagreement, they have provided comprehensive arguments that allow us to accept the manuscript in its present form

Reviewer #2: Thank you very much for revising your manuscript. The manuscript requires following points to be addressed in another round of revision:

1. The authors have made many changes according to the specific comments. However, I was unable to see any those changes highlighted or in different font color. This made the evaluation of revisions in revised manuscript difficult to follow. Any changes made should have been highlighted or changed font color even if suggested or incorporated by the reviewer.

2. The language is consistently poor. From the authors' responses, it seems that they did not consult any native English speaking colleague or a language editing service. The authors should consider to improve the language to make it for better understanding of readers.

3. The authors have been unable to convince why there was not a comparison with a representative laying hen breed as a control group. Also, most of the breeds they evaluated were hybrids (e.g., Araucana, Kurohisui, and Boris Brown). The authors exert that Boris Brown (a hybrid cross) is better for making a comparison. Even if the authors want to compare other breeds with BOR, it should be declared somewhere in the manuscript.

4. The authors have been unable to remove the discussion part related to floor rearing system. The authors should understand that floor rearing system was not used as a factor. Since the floor rearing system was used across the groups (breeds in this case), the discussion related to floor rearing system becomes irrelevant and should be deleted.

7. PLOS authors have the option to publish the peer review history of their article (what does this mean?). If published, this will include your full peer review and any attached files.

Reviewer #1: No

Reviewer #2: No

---

## [Author Response · Author response to Decision Letter 1]

26 Aug 2021

6. Review Comments to the Author

Reviewer #1: The authors made the suggested revisions to the manuscript. In addition, in situations of disagreement, they have provided comprehensive arguments that allow us to accept the manuscript in its present form

Reviewer #2: Thank you very much for revising your manuscript. The manuscript requires following points to be addressed in another round of revision:

1. The authors have made many changes according to the specific comments. However, I was unable to see any those changes highlighted or in different font color. This made the evaluation of revisions in revised manuscript difficult to follow. Any changes made should have been highlighted or changed font color even if suggested or incorporated by the reviewer.

[Authors’ response] Editorial Manager required both clean version and revised manuscript with track changes (changed font color by red). So, we have submitted the two versions of first revised manuscripts. However, we have not expressed all the things which you indicated comprehensive grammatical suggestions with track changes. Therefore, we indicated all the first round revised points by red letters and the second round points by blue letters. If you evaluate all the things throughout the manuscript, we are very happy. 

2. The language is consistently poor. From the authors' responses, it seems that they did not consult any native English speaking colleague or a language editing service. The authors should consider to improve the language to make it for better understanding of readers.

[Authors’ response] We had our native English-speaking colleague help with revising our manuscript before submitting. Since there were so many grammar errors, we revised the manuscript entirely (blue letters). In addition, we could receive your comprehensive concrete advices using WORD track change function in the first round of revise. Your concrete advices make our manuscript better for potential readers. We really appreciate it. 

3. The authors have been unable to convince why there was not a comparison with a representative laying hen breed as a control group. Also, most of the breeds they evaluated were hybrids (e.g., Araucana, Kurohisui, and Boris Brown). The authors exert that Boris Brown (a hybrid cross) is better for making a comparison. Even if the authors want to compare other breeds with BOR, it should be declared somewhere in the manuscript.

[Authors’ response] Thank you very much. We described “Since BOR (GHEN Corporation, Japan) is the world’s brown egg layer (Hy-Line Brown; Hy-Line International, USA), we selected BOR as representative hen for the comparison.” (L 102-104). 

4. The authors have been unable to remove the discussion part related to floor rearing system. The authors should understand that floor rearing system was not used as a factor. Since the floor rearing system was used across the groups (breeds in this case), the discussion related to floor rearing system becomes irrelevant and should be deleted.

[Authors’ response] Thank you very much. We will analyze and report the effect of rearing systems on egg amino acids in the future. According to your suggestions, we deleted a whole paragraph in discussion from this paper. 

The deleted paragraph is shown below. 

We hypothesized that there are some different features in the free amino acid contents of yolk and albumen between floor and cage rearing systems. Our previous report indicated UKO > NGY in both yolk and albumen amino acid contents under cage rearing system [17]. However, this study revealed NGY > UKO in both yolk and albumen amino acid traits under floor rearing system. Moreover, Brown layer (BOR) produced the lowest concentrations of albumen amino acids in this study. Although BOR is frequently used for egg production by floor reared hens in Japan, it may be another choice of genotypes for producing amino acid enriched eggs under floor rearing system.

---

## [Decision Letter · Decision Letter 2]

29 Sep 2021

Genetic effect on free amino acid contents of egg yolk and albumen using five different chicken genotypes under floor rearing system

PONE-D-21-14031R2

Dear Dr. Tatsuhiko Goto,

We’re pleased to inform you that your manuscript has been judged scientifically suitable for publication and will be formally accepted for publication once it meets all outstanding technical requirements.

Kind regards,

Ewa Tomaszewska, DVM Ph.D

Academic Editor

PLOS ONE

Additional Editor Comments (optional):

Reviewers' comments:

Reviewer's Responses to Questions

**Comments to the Author**

1. If the authors have adequately addressed your comments raised in a previous round of review and you feel that this manuscript is now acceptable for publication, you may indicate that here to bypass the “Comments to the Author” section, enter your conflict of interest statement in the “Confidential to Editor” section, and submit your "Accept" recommendation.

Reviewer #1: All comments have been addressed

Reviewer #2: All comments have been addressed

2. Is the manuscript technically sound, and do the data support the conclusions?

Reviewer #1: (No Response)

Reviewer #2: Yes

3. Has the statistical analysis been performed appropriately and rigorously? 

Reviewer #1: Yes

Reviewer #2: (No Response)

4. Have the authors made all data underlying the findings in their manuscript fully available?

Reviewer #1: Yes

Reviewer #2: Yes

5. Is the manuscript presented in an intelligible fashion and written in standard English?

Reviewer #1: Yes

Reviewer #2: Yes

6. Review Comments to the Author

Reviewer #1: (No Response)

Reviewer #2: (No Response)

7. PLOS authors have the option to publish the peer review history of their article (what does this mean?). If published, this will include your full peer review and any attached files.

Reviewer #1: No

Reviewer #2: No

---

## [Editor Report · Acceptance letter]

1 Oct 2021

PONE-D-21-14031R2 

Genetic effect on free amino acid contents of egg yolk and albumen using five different chicken genotypes under floor rearing system 

Dear Dr. Goto:

I'm pleased to inform you that your manuscript has been deemed suitable for publication in PLOS ONE. Congratulations! Your manuscript is now with our production department. 

Kind regards, 

on behalf of

Prof. Dr. Ewa Tomaszewska 

Academic Editor

PLOS ONE